# Comparison of the Clinical Process and Outcomes in Patients after Coronavirus Infection 2019 Outbreak

**DOI:** 10.3390/medicina57101086

**Published:** 2021-10-11

**Authors:** Sung-Jin Bae, Ho-Sub Chung, Myeong Namgung, Yoon-Hee Choi, Jin-Hong Min, Dong-Hoon Lee

**Affiliations:** 1Department of Emergency Medicine, College of Medicine, Chung-Ang University, Seoul 06973, Korea; uzimuz85@gmail.com (S.-J.B.); hoshap@hanmail.net (H.-S.C.); myeong15180@caumc.or.kr (M.N.); 2Department of Emergency Medicine, Ewha Womans University Mokdong Hospital, College of Medicine, Ewha Womans University, Seoul 07985, Korea; unii@ewha.ac.kr; 3Department of Emergency Medicine, College of Medicine, Chungnam National University, Daejeon 35015, Korea; laphir@cnu.ac.kr

**Keywords:** COVID-19, emergency medicine, delayed treatments, emergency department, length of stay

## Abstract

*Background and Objectives*: The coronavirus infection 2019 (COVID-19) pandemic has affected emergency department (ED) management. Its viral transmission necessitates the use of isolation rooms and personal protective equipment for treating suspected patients, such as those with fever. This delays the time until the first encounter with the patients, thereby increasing the length of stay (LOS) in the ED. We aimed to compare delays in the ED LOS and clinical processes between the COVID-19 period and pre-COVID-19 period. Moreover, we intended to evaluate if the aforementioned delay affected patient outcomes. *Materials and Methods*: We conducted a single-center, retrospective study in Korea. Patients with fever were compared between the “COVID-19 period” from March 2020 to August 2020 and the “pre-COVID-19 period” from March 2019 to September 2019. We compared the overall ED LOS and individual time variable, including initial diagnostic tests (laboratory tests, radiography), specific diagnostic test (computed tomography), and treatment processes (antibiotics). A logistic regression analysis was conducted to identify the association between hospital admission and patient data. *Results*: We enrolled 931 and 749 patients during pre- and COVID-19 periods, respectively. Patients with fever remained in the ED for a longer duration during the COVID-19 period (pre-COVID-19:207.7 ± 102.7 min vs. during COVID-19: 223.5 ± 119.4 min, *p* = 0.004). The total time for performing laboratory tests and radiography displayed significant differences between the two periods, particularly from the time of patient arrival in the ED to the time of issuing the order. The time until antibiotic administration was delayed in the COVID-19 period (pre-COVID-19:195.8 ± 103.3 min vs. during COVID-19: 216.9 ± 108.4 min, *p* = 0.003). The logistic regression analysis for hospital admission identified ED LOS as an independent factor in both periods. *Conclusions*: The delay until encountering patients with fever resulted in longer ED LOS during the COVID-19 period; however, it possibly did not increase the hospital admission rates.

## 1. Introduction

Coronavirus infection 2019 (COVID-19) is a disease caused by the new coronavirus [1]. Despite global efforts, COVID-19 has spread worldwide, with an ever increasing number of patients and deaths [2]. Eventually, the World Health Organization issued a pandemic declaration for COVID-19 on 11 March 2020 [3]. In South Korea, the first case of COVID-19 was reported on 20 January 2020, followed by an explosive increase in February 2020 [4]. Thus, the Korean government raised the COVID-19 the Infectious Disease crisis alert level to “serious” on 23 February 2020 [5]. The government restricted the operation of facilities at risk of mass infection (e.g., sports events, schools, religious facilities, etc.). As a result, the number of confirmed cases seemed to be decreasing; however, when the restrictions were eased, the number of confirmed cases increased again.

Changes in the healthcare system were necessary to prepare for the novel pandemic viral disease. The government allocated COVID-19-designated hospitals to provide isolation beds for patients, and each hospital established isolation guidelines for patient care. The medical staff were required to wear certified personal protective equipment (PPE) to prevent exposure and infection. The U.S. Centers for Disease Control and Prevention recommends gloves, gowns, respiratory protection, and eye protection as standardized PPE medical personnel to control COVID-19 infection [6].

According to initial cases, more than half of the patients had symptoms such as fever, fatigue, and dry cough [7]. Further cohort studies identified myalgia, dyspnea, and anorexia in those positive for COVID-19 [8,9]. Confirmed cases of patients without symptoms increased over time [10]. Several subsequent cohort studies have also identified other symptoms; however, fever has been identified as a chief complaint [11,12]. In addition, fever is one of the most common symptoms in the emergency department (ED). It is prevalent in 5%, 15%, and 40% of visits in adults, the elderly, and children, respectively [13]. The use of isolation rooms and PPE by the medical staff resulted in spending more time on patients with fever. Moreover, the isolation room might be unavailable for a patient with fever visiting the ED because of the facility being occupied by other patients, thus delaying the treatment.

We aimed to determine the time delay experienced by patients with fever who visited the ED during the COVID-19 pandemic and the corresponding stage during the clinical process. We also aimed to investigate variations in the disposition of patients before and during the pandemic.

## 2. Materials and Methods

### 2.1. Study Design and Population

This retrospective study comprised patients visiting the ED at a tertiary university hospital, located in the capital of South Korea with a population of about 9 million. About 30,000 to 40,000 patients visited the ED annually. Their medical data were collected from two groups, considering seasonal variations. First, the “COVID-19 period” was defined as patient data collected from March 2020 to August 2020, following the change in the Infectious Disease crisis alert to “serious” level. In contrast, the “pre-COVID-19 period” was defined as data collected from March 2019 to August 2019. The inclusion criteria were as follows: (1) patients aged ≥18 years who visited the ED, (2) with body temperature > 37.5 °C during the ED visit. The exclusion criteria were as follows: (1) patients who presented as dead-on-arrival; (2) ED visits for non-medical purposes; (3) patients with unknown prognosis because of a transfer to another hospital or discharge against medical advice; (4) patients aged <18 years; and (5) trauma-related ED visits (Figure 1). This study was approved by the institutional review board of our hospital, and the need for informed patient consent was waived (IRB No. 2107-034-19376).

### 2.2. Data Collection and Outcome Measurement

The study variables included patient demographics, initial vital signs, mental status, the Korean Triage and Acuity Scale (KTAS) score, diagnosis at ED discharge, final diagnosis at hospital discharge, the length of ED stay, disposition in the ED, the length of hospital stay, and in-hospital mortality. We used the International Classification of Diseases (ICD-10) code to classify the diagnosis. The KTAS was developed for use in Korean EDs as a triage tool. It categorizes patients from 1 to 5 based on a severity scale, with 1 being the most severe. To compare the delay in ED management, we investigated the time variables of laboratory tests, radiography, computed tomography (CT), and antibiotics. We recorded the time point for each variable, including the time to order and perform. We defined them as follows. (1) Door to order: from the time of patient arrival in the ED to the time of issuing the order. (2) Order to perform: from the time the order was issued to the time it was performed. (3) Door to perform: from the time of patient arrival in the ED to the time of implementing the order. The primary outcome was the time delay in the treatment process during COVID-19 period, compared to that during the pre-COVID-19 period. The secondary outcomes comprised the admission rate and in-hospital mortality.

### 2.3. Statistical Analyses

Continuous variables are reported as means with standard deviations (SD), and categorical variables are reported as counts (percentages). To compare the distribution of characteristics between each period, we performed a chi-squared test for the categorical variables, including sex, mental status, KTAS, disposition, and antibiotics. We performed the independent *t*-test or Mann–Whitney U-test for the continuous variables, including the age, vital signs, and time variables. To identify the independent prognostic factor for each patient group, we performed a multivariate logistic regression analysis of the variables that were statistically significant in the univariate logistic regression. Moreover, we obtained the adjusted odds ratios (ORs) and 95% confidence intervals (CIs) from the multivariate analyses. The significance level was set as *p* < 0.05. All statistical analyses were conducted using SPSS 26.0 (SPSS Inc., Chicago, IL, USA).

## 3. Results

### 3.1. Baseline Characteristics

From March 2019 to August 2019, 23,142 patients visited the ED. Of these patients, 931 were enrolled in the study (the pre-COVID-19 period). Between March 2020 and August 2020, 15,761 patients visited the ED, of whom 749 were eventually enrolled (the COVID-19 period) (Figure 1). Table 1 summarizes the baseline characteristics for both groups. The mean age of both groups was 51.7 ± 23.5 years and 53.9 ± 23.3 years, respectively (*p* = 0.048). The pre- and COVID-19 groups comprised 47.6% and 50.1% men, respectively (*p* = 0.311). Moreover, 50.4% and 41.8% patients demonstrated 1, 2, or 3 points of KTAS during the pre- and COVID-19 periods, respectively (*p* < 0.001). KTAS scores of 3 and 4 accounted for the largest number of patients in both periods. Similarly, as a severity indicator, 2.5% and 3.5% patients demonstrated quick a Sequential (Sepsis-related) Organ Failure Assessment score ≥ 2 in the pre- and COVID-19 groups, respectively, without a significant difference (*p* = 0.226).

### 3.2. The Distribution of Diagnosis According to the ICD-10 Classification

Table 2 summarizes the distribution of diagnoses according to the ICD-10 classification. The diagnosis of a patient discharged from the ED was as follows: in the pre-COVID-19 period, X-disease of the respiratory system code accounted for the largest percentage with 316 patients (33.9%). In the COVID-19 period, XVII-symptoms, signs and abnormal clinical and laboratory findings, not elsewhere classified had the largest number of patients (24%), followed by X-disease of the respiratory system (18.7%), which had the largest prevalence in the pre-COVID-19 period. For the final diagnosis of patients after admission, X-diseases of the respiratory system accounted for the largest proportion in the pre-COVID-19 period (23.5%), compared to XI-diseases of the digestive system that accounted for the largest proportion in the COVID-19 period (21.3%).

### 3.3. Comparison of Time Variables in the Clinical Processes by the Pre- and COVID-19 Period

Table 3 summarizes the time variables recorded during the clinical process for both groups. The length of stay in the ED was longer during the COVID-19 period than during the pre-COVID-19 period (223.5 ± 119.4 min vs. 207.7 ± 102.7 min) (*p* = 0.004). The pre-COVID-19 period displayed a statistically significant difference in the total time of performing laboratory tests from the ED visit than the COVID-19 period (82.9 ± 26.7 min vs. 87.2 ± 39.2 min) (*p* = 0.018). The total times for performing radiography were 86.3 ± 50.5 min and 92.2 ± 57.6 min for the pre- and COVID-19 periods, respectively (*p* = 0.047). The difference in the overall time for performing CT scans was statistically insignificant between the groups (*p* = 0.933). The door to order, and not the order to perform, displayed a statistically significant difference in the laboratory tests and radiography. The times until antibiotic administration were 195.8 ± 103.3 min and 216.9 ± 108.4 min for the pre- and COVID-19 periods, respectively, which were further delayed (*p* = 0.003).

### 3.4. Antibiotics Administered during the Pre- and COVID-19 Period

There was no statistically significant difference in antibiotic treatment during pre-and COVID-19 periods (51.8% vs. 54.3%) (*p* = 0.295). Quinolone was the most commonly implicated intravenous antibiotic product in both groups, accounting for 44.4% and 34.9% in pre-and COVID-19 groups, respectively. Third-generation cephalosporin was the second most commonly implicated antibiotic product in pre-and COVID-19 groups (22.6% vs. 27.3%), followed by penicillin and fourth-generation cephalosporin (Table 4).

### 3.5. Predicting Factors for Hospital Admission in Pre- and COVID-19 Periods

Table 5 outlines the multivariate logistic regression analysis of hospital admission during pre- and COVID-19 periods. In the pre-COVID-19 period, the age (OR 1.040; 95% CI 1.025–1.054; *p* < 0.001), male sex (OR 2.011; 95% CI 1.197–3.376; *p* = 0.008), KTAS triage category ≥3 (OR 2.808; 95% CI 1.646–4.791; *p* < 0.001), the length of ED stay (OR 1.007; 95% CI 1.004–1.011; *p* < 0.001), and radiography (door to perform) (OR 1.006; 95% CI 1.000–1.011; *p* = 0.034) were identified as the independent factors for predicting admission. In COVID-19 period, the age (OR 1.036; 95% CI 1.022–1.050; *p* < 0.001), respiratory rate (OR 1.318; 95% CI 1.122–1.548; *p* = 0.001), KTAS triage category ≥3 (OR 2.098; 95% CI 1.205–3.654; *p* = 0.009), and the length of ED stay (OR 1.003; 95% CI 1.000–1.006; *p* = 0.037) were considered independent factors for predicting admission.

## 4. Discussion

This study compared the length of ED stay and the time interval during diagnosis and treatment between pre- and COVID-19 periods. The length of stay in the ED was longer at 223.5 min during the COVID-19 period compared to 207.7 min during the pre-COVID-19 period (*p* = 0.004, Table 3). Previous studies have reported on the contagious nature of COVID-19 and high rates of transmission in humans [14,15,16]. Emergency physicians should be aware of the possibility of COVID-19 for patients presenting with fever. This warrants more time and effort for patient management. This can be attributed to the need for patient admission to isolation rooms and use of PPE by all medical personnel. The number of patients visiting the ED during the COVID-19 period decreased by 31.9%. Of them, the number of patients with fever decreased by 19.5% (931 during the pre-COVID-19 vs. 749 during the COVID-19 period). This trend was similar to that reported in previous studies [17,18]. The major contributing factor for the decrease in ED visits during the pandemic was probably the fear of patients being exposed to COVID-19. This fear of exposure to COVID-19 in hospitals was also reported in patients with stroke and ST-elevation myocardial infarction [19,20].

Researchers reported on an outbreak of Middle East Respiratory Syndrome (MERS), which became a serious infectious disease worldwide. In South Korea, 16,752 presumed MERS cases were reported in 2015, with 38 deaths [21,22]. Based on the MERS outbreak experience, the Korean government implemented countermeasures for reforming the prevention system [23,24]. One hundred and sixty-five general hospitals and 20 tertiary hospitals have been designated as isolation rooms and specialized hospitals for local infectious diseases. In December 2015, a law was enacted requiring the establishment of isolation rooms at Regional Emergency Medical Centers. More isolation rooms have since been added to the emergency medical center in preparation for another infection [24]. Despite these preparations, the transmission of COVID-19 was rapid, thereby causing difficulty in containing the disease. Thus, a larger number of possibly infected patients visited the ED.

Prolonged stay in the ED owing to time delay can cause ED overcrowding, which, in turn, can burden both patients and medical staff. The length of ED stay for patients with fever was longer in the COVID-19 period than in the pre-COVID-19 period. We further examined the time variable that could affect the ED stay duration. Physicians usually performed history taking and physical examinations during their first encounter with patients. On suspecting infectious disease based on this first encounter, physicians ordered laboratory tests, including complete blood count (CBC) and simple radiography. This was followed by CT to confirm specific abnormalities or suspected conditions in these initial tests. The time from the door to order of CBC and chest radiography could possibly reflect the first encounter with patients, and the time for CT order could denote repeated encounters in ED. The time from the order to performing these tests could reveal the delay in management progress in the ED. Furthermore, we compared the time of antibiotic administration to investigate the delay in treatment. The time of the door to order of CBC and radiography was delayed in the COVID-19 period; however, there was no difference in the time from the order to performing tests. In other words, physicians took longer for their first encounter with patients during the COVID-19 period. This can be attributed to the need for isolation rooms and the use of PPE.

Rapid and accurate antibiotic treatment is essential to avoid severe complications and reduce mortality in infected patients [25,26]. There was no significant difference in the prescription rates (*p* = 0.295) or the type of antibiotics administered (*p* = 0.179). However, the time from the visit to antibiotic administration during the COVID-19 period (216.9 ± 108.4 min) was longer than that during the pre-COVID-period (195.8 ± 103.3 min). The delay may be attributed to the time-consuming process of the first physician’s encounter with patients or the use of PPE by nurses for skin tests or antibiotic administration. Researchers have conducted studies on ED LOS during the COVID-19 period. Lucero et al. [27] reported on increased length of ED stay during the COVID-19 period, and attributed it to hospital overcrowding because of an increase in the rate of COVID-19 admission. Moreover, Chen et al. [28] reported that the length of ED stay increased during the pandemic, and expected an increase in the proportion of critical patients to be one of the causes.

The hospital admission rate was higher during the COVID-19 period (43.8%), than during the pre-COVID-19 period (35.7%) (*p* = 0.001, Table 1). However, there were no statistically significant differences in the in-hospital mortality (*p* = 0.453) or intensive care unit admission rates (*p* = 0.961). The multivariate logistic analysis identified longer ED LOS as an independent predictive factor in both pre- and COVID-19 periods (OR 1.007, 95% CI 1.004–1.011 vs. OR 1.003, 95% CI 1.000–1.006, Table 5). However, the delay during the first encounter by physicians in the univariate analysis was statistically unrelated to hospital admission. The length of ED stays during the COVID-19 period was longer because of delays until the first encounter with patients; however, it did not increase the hospital admission rate.

This study has some limitations. First, the study had a retrospective design and was conducted at a single hospital, which may have introduced selection bias. Second, the capacity and protocol for the management of infectious disease may have varied across hospitals. Depending on the location and size of the hospital, the number and characteristics of patients visiting may vary. Additionally, the number of isolation rooms in the ED varies depending on the hospital. However, it is mandatory for medical staff to wear PPE and use isolation rooms for the treatment of patients with fever. Therefore, we think the problem caused by the delay in patient care will appear in common. Third, patient characteristics may change depending on the timing of the epidemic. However, the COVID-19 pandemic is ongoing; therefore, we could not be categorized it into early, mid, and late sections. Last, in this study, we analyzed all patients with fever who visited the ED. Therefore, we did not compare clinical outcomes that reflect the severity of each disease.

## 5. Conclusions

In conclusion, the time of stay in the ED was longer during the COVID-19 period than during the pre-COVID-19 period. Moreover, physicians took longer for their first encounter with patients in the COVID-19 period. Despite higher hospital admission rates in the COVID-19 period, it was not significantly related to the time of first encounter.

## Figures and Tables

**Figure 1 medicina-57-01086-f001:**
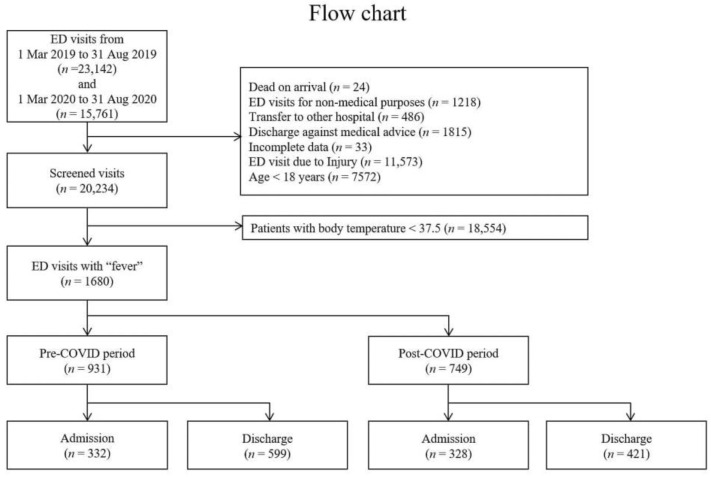
Study design and patient enrollment.

**Table 1 medicina-57-01086-t001:** The characteristics of adult patients by the pre- and COVID period.

Variable	Pre-COVID-19 Period (*n* = 931)	COVID-19 Period (*n* = 749)	*p*-Value
Age (y) ^†^	51.7 ± 23.5	53.9 ± 23.3	0.048
Sex ^‡^			0.311
Male	443 (47.6)	375 (50.1)	
Female	488 (52.4)	374 (49.9)	
Vital sign ^†^			
Systolic blood pressure (mmHg)	130.4 ± 22.7	135.2 ± 26.2	<0.001
Diastolic blood pressure (mmHg)	75.5 ± 13.1	77.7 ± 13.5	0.001
Pulse rate (beats/min)	101.1 ± 18.1	101 ± 18.4	0.880
Respiratory rate (breath/min)	20.5 ± 2.2	19.8 ± 1.9	<0.001
Body temperature (°C)	38.2 ± 0.6	38.1 ± 0.5	0.004
Mental status ^‡^			0.389
Alert	906 (97.3)	721 (96.3)	
Verbal response	14 (1.5)	18 (2.4)	
Painful response	11 (1.2)	10 (1.3)	
Unresponsive	0	0	
KTAS Triage category ^‡^			0.002
Level 1 Resuscitation	1 (0.1)	2 (0.3)	
Level 2 Emergent	30 (3.2)	32 (4.3)	
Level 3 Urgent	438 (47.0)	279 (37.2)	
Level 4 Less urgent	453 (48.7)	427 (57.0)	
Level 5 Non urgent	9 (1.0)	9 (1.2)	
qSOFA ≥2 ^‡^	23 (2.5)	26 (3.5)	0.226
ED Disposition ^‡^			0.001
Discharge ^‡^	599 (64.3)	421 (56.2)	
Admission ^‡^	332 (35.7)	328 (43.8)	
ICU admission ^‡^	47 (14.2)	46 (14.0)	0.961
Hospital LOS (d) ^§^	10 (6.25–18.00)	10.50 (6.00–20.00)	0.807
In-hospital mortality ^‡^	25 (7.5)	30 (9.1)	0.453

^†^ The values are given as mean ± standard deviation. ^‡^ The values are given as number (%). ^§^ The values are given as median (interquartile range). COVID-19: coronavirus disease-2019; ICU: intensive care unit; KTAS: Korean Triage and Acuity Scale; LOS: length of stay; qSOFA: quick Sequential (Sepsis-related) Organ Failure Assessment.

**Table 2 medicina-57-01086-t002:** The distribution of diagnosis according to the ICD-10 by the Pre- and COVID period.

Diagnosis at Emergency Department Discharge			
Pre-COVID-19 Period		COVID-19 Period	
(*n* = 941)		(*n* = 759)	
X Disease of the respiratory system	316 (33.9)	XVIII Symptoms, signs and abnormal clinical and laboratory findings, not elsewhere classified	180 (24.0)
I Certain infectious and parasitic disease	166 (17.8)	X Disease of the respiratory system	140 (18.7)
XIV Diseases of the genitourinary system	134 (14.4)	I Certain infectious and parasitic disease	110 (14.7)
XVIII Symptoms, signs and abnormal clinical and laboratory findings, not elsewhere classified	127 (13.6)	XIV Diseases of the genitourinary system	101 (13.5)
XI Diseases of the digestive system	85 (9.1)	XI Diseases of the digestive system	74 (9.9)
II Neoplasm	26 (2.8)	II Neoplasm	49 (6.5)
IX Diseases of the circulatory system	20 (2.1)	IX Diseases of the circulatory system	26 (3.5)
XII Diseases of the skin and subcutaneous tissue	17 (1.8)	XII Diseases of the skin and subcutaneous tissue	17 (2.3)
III Diseases of the blood and blood-forming organs and certain disorders involving the immune mechanism	13 (1.4)	XIII Diseases of the musculoskeletal system and connective tissue	14 (1.9)
VI Diseases of the nervous system	8 (0.9)	XXI Factors influencing health status and contact with health services	8 (1.1)
IV Endocrine, nutritional and metabolic diseases	5 (0.5)	III Diseases of the blood and blood-forming organs and certain disorders involving the immune mechanism	5 (0.7)
V Mental and behavioral disorders	4 (0.4)	VI Diseases of the nervous system	5 (0.7)
XIX Injury, poisoning and certain other consequences of external causes	4 (0.4)	IV Endocrine, nutritional and metabolic diseases	5 (0.7)
XIII Diseases of the musculoskeletal system and connective tissue	3 (0.3)	XIX Injury, poisoning and certain other consequences of external causes	4 (0.5)
XXI Factors influencing health status and contact with health services	3 (0.3)	V Mental and behavioral disorders	3 (0.4)
		VIII Diseases of the ear and mastoid process	3 (0.4)
		VII Diseases of the eye and adnexa	2 (0.3)
		XV Pregnancy, childbirth and the puerperium	2 (0.3)
		XVII Congenital malformations, deformations and chromosomal abnormalities	1 (0.1)
**Diagnosis at Inpatient Discharge**			
**Pre-COVID-19 Period**		**COVID-19 Period**	
**(*n* = 332)**		**(*n* = 328)**	
X Diseases of the respiratory system	78 (23.5)	XI Diseases of the digestive system	70 (21.3)
XI Diseases of the digestive system	72 (21.7)	X Diseases of the respiratory system	67 (20.4)
XIV Diseases of the genitourinary system	62 (18.7)	XIV Diseases of the genitourinary system	51 (15.5)
II Neoplasms	39 (11.7)	II Neoplasms	35 (10.7)
I Certain infectious and parasitic diseases	23 (6.9)	IX Diseases of the circulatory system	25 (7.6)
IX Diseases of the circulatory system	17 (5.1)	XVIII Symptoms, signs and abnormal clinical and laboratory findings, not elsewhere classified	21 (6.4)
III Diseases of the blood and blood-forming organs and certain disorders involving the immune mechanism	8 (2.4)	I Certain infectious and parasitic diseases	18 (5.5)
XVIII Symptoms, signs and abnormal clinical and laboratory findings, not elsewhere classified	8 (2.4)	III Diseases of the blood and blood-forming organs and certain disorders involving the immune mechanism	11 (3.4)
IV Endocrine, nutritional and metabolic diseases	6 (1.8)	XIX Injury, poisoning and certain other consequences of external causes	7 (2.1)
XIX Injury, poisoning and certain other consequences of external causes	6 (1.8)	VI Diseases of the nervous system	5 (1.5)
VI Diseases of the nervous system	4 (1.2)	XII Diseases of the skin and subcutaneous tissue	5 (1.5)
XIII Diseases of the musculoskeletal system and connective tissue	4 (1.2)	XIII Diseases of the musculoskeletal system and connective tissue	4 (1.2)
XII Diseases of the skin and subcutaneous tissue	3 (0.9)	XV Pregnancy, childbirth and the puerperium	3 (0.9)
XXI Factors influencing health status and contact with health services	2 (0.6)	XXI Factors influencing health status and contact with health services	2 (0.6)
		IV Endocrine, nutritional and metabolic diseases	1 (0.3)
		VII Diseases of the eye and adnexa	1 (0.3)
		VIII Diseases of the ear and mastoid process	1 (0.3)
		XVII Congenital malformations, deformations and chromosomal abnormalities	1 (0.3)

All values are given as number (%). COVID-19: coronavirus disease-2019; ICD: International Classification of Diseases.

**Table 3 medicina-57-01086-t003:** The time variables of clinical processes by the Pre- and COVID period.

Variable	Pre-COVID-19 Period (*n* = 931)	COVID-19 Period (*n* = 749)	*p*-Value
ED LOS (min) ^a^	207.7 ± 102.7	223.5 ± 119.4	0.004
Laboratory tests (CBC) ^†^			
Door to Order (min) ^a^	50.8 ± 16.3	54 ± 23.2	0.003
Order to Perform (min) ^a^	32.2 ± 12.3	33.2 ± 17.8	0.212
Door to Perform (min)	82.9 ± 26.7	87.2 ± 39.2	0.018
Radiography ^‡^			
Door to Order (min) ^a^	54.3 ± 30	59.4 ± 38.8	0.009
Order to Perform (min) ^a^	32 ± 24.5	32.9 ± 25.5	0.540
Door to Perform (min)	86.3 ± 50.5	92.2 ± 57.6	0.047
Computed tomography ^§^			
Door to Order (min) ^a^	167.5 ± 70.6	165 ± 63	0.726
Order to Perform (min) ^a^	40.5 ± 27.8	43.8 ± 35.2	0.327
Door to Perform (min)	207.9 ± 88.3	208.7 ± 89.6	0.933
Door to Antibiotics (min) ^¶,a^	195.8 ± 103.3	216.9 ± 108.4	0.003

^a^ The values are presented as mean ± standard deviation. ^†^ Laboratory tests were performed in 871 and 649 patients in pre- and COVID period, respectively. ^‡^ Radiography was performed in 717 and 606 patients in pre- and COVID period, respectively. ^§^ Computed tomography tests were performed in 185 and 168 patients in pre- and COVID period, respectively. ^¶^ Antibiotics were treated in 482 and 407 patients in pre- and COVID period, respectively. CBC: complete blood count; COVID-19: coronavirus disease-2019; ED: emergency department; LOS: length of hospital stay.

**Table 4 medicina-57-01086-t004:** The distribution of antibiotics administered to patients by the Pre- and COVID period.

Variable	Pre-COVID-19 Period	COVID-19 Period	*p*-Value
	(*n* = 931)	(*n* = 749)	
Antibiotics			0.295
No	449 (48.2)	342 (45.7)	
Yes	482 (51.8)	407 (54.3)	
			0.179
1st cephalosporin	11 (2.3)	13 (3.2)	
2nd cephalosporin	6 (1.2)	2 (0.5)	
3rd cephalosporin	109 (22.6)	111 (27.3)	
4th cephalosporin	47 (9.8)	39 (9.6)	
Aminoglycosides	1 (0.2)	0	
Carbapenem	33 (6.8)	34 (8.4)	
Penicillin	59 (12.2)	64 (15.7)	
Quinolone	214 (44.4)	142 (34.9)	
Vancomycin	2 (0.4)	2 (0.5)	

All values are given as number (%). Abbreviations: COVID-19: coronavirus disease-2019.

**Table 5 medicina-57-01086-t005:** Multivariate logistic regression analysis of admission predictors.

**Pre-COVID-19 Period**			
**Variable**	**OR**	**B**	***p*-Value**
Age (years)	1.040 (1.025–1.054)	0.039	<0.001
Sex; Male	2.011 (1.197–3.376)	0.698	0.008
KTAS triage category ≤ 3	2.808 (1.646–4.791)	1.033	<0.001
ED LOS (min)	1.007 (1.004–1.011)	0.007	<0.001
Radiography			
Door to Perform (min)	1.006 (1.000–1.011)	0.006	0.034
**COVID-19 Period**			
**Variable**	**OR**	**B**	***p*-Value**
Age (years)	1.036 (1.022–1.050)	0.035	<0.001
Respiratory rate (breath/min)	1.318 (1.122–1.548)	0.276	0.001
KTAS triage category ≤ 3	2.098 (1.205–3.654)	0.741	0.009
ED LOS (min)	1.003 (1.000–1.006)	0.003	0.037

OR: odds ratio, B: regression coefficient. Boldface type indicates statistical significance (*p* < 0.05). Abbreviations: COVID-19: coronavirus disease-2019; ED: emergency department; KTAS: Korean Triage and Acuity Scale; LOS: length of stay.

## Data Availability

The data presented in this study are available on request from the corresponding author. The data are not publicly available due to privacy or ethical restrictions.

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
