# Peer review of "Comparison of the Clinical Process and Outcomes in Patients after Coronavirus Infection 2019 Outbreak"

_medicina, 2021, doi:10.3390/medicina57101086_

Round 1

Reviewer 1 Report

Thank you for the interesting study. Even though the study has a retrospective design it points towards certain delays in everyday work in the emergency department. I have certain comments that could improve the quality of the manuscript. 

The introduction could be shortened and more focused. Certain typos should be corrected. In the first sentence of the introduction (Coronavirus infection 2019 (COVID-19) is a disease caused by the new coronavirus 42 syndrome [1]. ) the word syndrome should be replaced by virus.

Methods/Results. Could you elaborate on the category of ED visits for the non-medical purpose which comprises more than 1000 patients? Maybe those patients should not even be listed in the flowchart.

Discussion: Please further elaborate on possible limitations of the study especially regarding the second limitation. How could different protocols in different hospitals influence the results of the current study which was conducted in only one hospital? 

Plans for further research should be made. Also whether these results will influence the work in the ED. Maybe certain protocols could be modified. Please elaborate.

Author Response

Reviewer 1’s comments

Thank you for reviewing for our manuscript. I wrote down our thoughts on your opinion below.

           The introduction could be shortened and more focused. Certain typos should be corrected. In the first sentence of the introduction (Coronavirus infection 2019 (COVID-19) is a disease caused by the new coronavirus 42 syndrome [1].) the word syndrome should be replaced by virus.

Response: According to your recommendations, we have revised the ‘Introduction’ section. The details of the changes are as below.

Original sentence (Introduction; page 1, paragraph 1)

Coronavirus infection 2019 (COVID-19) is a disease caused by the new coronavirus syndrome [1].

Revised sentence (Introduction; page 1, paragraph 1)

Coronavirus infection 2019 (COVID-19) is a disease caused by the new coronavirus [1].

Also, we revised the introduction briefly, avoiding repeated expressions.

Original sentence (Introduction; page 2, paragraph 3)

According to initial cases, more than half of the patients had symptoms, such as fever, fatigue, and dry cough [7]. Further cohort studies identified myalgia, dyspnea, and anorexia in those positive for COVID-19 [8, 9]. Confirmed cases of patients without symptoms increased over time [10]. Several subsequent cohort studies have also identified other symptoms; however, fever has been identified as a chief complaint [11, 12]. In addition, fever is one of the most common symptoms in the emergency department (ED). It is prevalent in 5%, 15%, and 40% of visits in adults, the elderly, and children, respectively [13]. This necessitated isolation rooms for the treatment of patients with fever who visited the ED. Moreover, the medical staff in the ED had to wear PPE. The use of isolation rooms and PPE by the medical staff resulted in spending more time on patients with fever. The isolation room might be unavailable for a patient with fever visiting the ED because of the facility being occupied by other patients, thus delaying the treatment.

Revised sentence (Introduction; page 2, paragraph 3)

According to initial cases, more than half of the patients had symptoms, such as fever, fatigue, and dry cough [7]. Further cohort studies identified myalgia, dyspnea, and anorexia in those positive for COVID-19 [8, 9]. Confirmed cases of patients with-out symptoms increased over time [10]. Several subsequent cohort studies have also identified other symptoms; however, fever has been identified as a chief complaint [11, 12]. In addition, fever is one of the most common symptoms in the emergency department (ED). It is prevalent in 5%, 15%, and 40% of visits in adults, the elderly, and children, respectively [13]. The use of isolation rooms and PPE by the medical staff resulted in spending more time on patients with fever. Moreover, the isolation room might be unavailable for a patient with fever visiting the ED because of the facility being occupied by other patients, thus delaying the treatment.

           Methods/Results. Could you elaborate on the category of ED visits for the non-medical purpose which comprises more than 1000 patients? Maybe those patients should not even be listed in the flowchart.

Reply: Our emergency department provides various documents such as medical certificates when requested later for patients who have visited the ED. These visits are marked as ED visits for non-medical purposes in our EMR. It was 3 percent of total patients during the study period, thus, we judged that it was not a small number of patients. So, we included them in the flowchart.

           Discussion: Please further elaborate on possible limitations of the study especially regarding the second limitation. How could different protocols in different hospitals influence the results of the current study which was conducted in only one hospital?

Reply: Depending on the location and size of the hospital, the number and characteristics of patients visiting may vary. Also, the number of isolation rooms in the ED varies depending on the hospital. However, it is mandatory for medical staff to wear PPE and use isolation rooms for the treatment of patients with fever. Therefore, we think the problem caused by the delay in patient care will appear in common. Other hospital's medical staffs can find improvements by applying the cause of the delay in treatment found in our study to each hospital. We added further elaboration.

Original sentence (Discussion; page 10, paragraph 3)

Second, the capacity and protocol for the management of infectious disease may have varied across hospitals.

Revised sentence (Discussion; page 10, paragraph 3)

Second, the capacity and protocol for the management of infectious disease may have varied across hospitals. Depending on the location and size of the hospital, the number and characteristics of patients visiting may vary. Also, the number of isolation rooms in the ED varies depending on the hospital. However, it is mandatory for medical staff to wear PPE and use isolation rooms for the treatment of patients with fever. Therefore, we think the problem caused by the delay in patient care will appear in common.

           Plans for further research should be made. Also whether these results will influence the work in the ED. Maybe certain protocols could be modified. Please elaborate

Reply: Currently, medical staffs and nurses in the ED are managing both isolation room patients and non-isolation room patients. The wearing and undressing of PPE continue to be repeated. The resulting a longer time for their first encounter with patients was confirmed in our study. It is believed that the medical staff dedicated to the isolation room may be the solution. Further research may be necessary by including several hospitals and dividing the treatment process in more detail. In addition, as the vaccination rate increases, only a small number of fever patients may need to use the isolation room. If this is made into an additional variable and the study is conducted, the presence or absence of vaccination in the patient may be included in the treatment protocol.

Reviewer 2 Report

no comment

Author Response

Thank you for reviewing for our manuscript. 

Reviewer 3 Report

Well written presentation of a very interesting study conducted by the authors. Though retrospective in nature, thorough details and information are given for the population of interest (results' section). Good discussion with clear limitation note. It may trigger similar studies from other healthcare systems and different ED organisation.

Yet, since in "COVID 19" period , different mass health measures (e.g. "lockdowns")  were in place, please mention what kind of  mass health measures were valid during study COVID period (March -Aug 2020).Also , please add some notes about the hospital and the area/ population that is covering.These information are essential to the reader to understand the frame within the hospital was working during the two periods.

Author Response

Reviewer 3’s comments

Thank you for reviewing for our manuscript. I wrote down our thoughts on your opinion below.

           Yet, since in "COVID 19" period, different mass health measures (e.g. "lockdowns") were in place, please mention what kind of mass health measures were valid during study COVID period (March -Aug 2020).

Reply: During the study period, strengthening and easing social distancing were repeated according to the number of confirmed cases. When the level of infectious disease crisis alert was first raised to "serious" and restricted the operation of facilities at risk of mass infection (e.g., sports events, schools, religious facilities, etc.), the number of confirmed cases decreased. We added mention this in the manuscript.

Original sentence (Introduction; page 2, paragraph 1)

Thus, the Korean government raised the COVID-19 the Infectious Disease crisis alert level to “serious” on February 23, 2020 [5].

Revised sentence (Introduction; page 2, paragraph 1)

Thus, the Korean government raised the COVID-19 the Infectious Disease crisis alert level to “serious” on February 23, 2020 [5]. The government restricted the operation of facilities at risk of mass infection (e.g., sports events, schools, religious facilities, etc.). As a result, the number of confirmed cases seemed to be decreasing, however, when the restrictions were eased, the number of confirmed cases increased again.

           Also , please add some notes about the hospital and the area/ population that is covering. These information are essential to the reader to understand the frame within the hospital was working during the two periods.

Reply: The hospital where the study was conducted is located in the center of the capital of Korea, with a population of about 9 million. About 30,000 to 40,000 patients visit the ED annually. Additional explanations were added to the method section.

Original sentence (Materials and Methods; page 2, paragraph 5)

This retrospective study comprised patients visiting the ED at a tertiary university hospital, located in the capital of South Korea.

Revised sentence (Materials and Methods; page 2, paragraph 5)

This retrospective study comprised patients visiting the ED at a tertiary university hospital, located in the capital of South Korea with a population of about 9 million. About 30,000 to 40,000 patients visited the ED annually.
